# Development, Validation, and Measurement Invariance of the Body Image Bidimensional Assessment (BIBA) in Italian and Spanish Children and Early Adolescent Samples

**DOI:** 10.3390/ijerph20043626

**Published:** 2023-02-17

**Authors:** Cristina Segura-Garcia, Matteo Aloi, Elvira Anna Carbone, Filippo Antonio Staltari, Marianna Rania, Maria Cristina Papaianni, Mikel Vaquero-Solís, Miguel Ángel Tapia-Serrano, Pedro Antonio Sánchez-Miguel, Marco Tullio Liuzza

**Affiliations:** 1Department of Medical and Surgical Sciences, University Magna Graecia of Catanzaro, 88100 Catanzaro, Italy; 2Outpatient Unit for Clinical Research and Treatment of Eating Disorders, University Hospital Mater Domini, 88100 Catanzaro, Italy; 3Department of Clinical and Experimental Medicine, University of Messina, 98125 Messina, Italy; 4Department of Health Sciences, University Magna Graecia of Catanzaro, 88100 Catanzaro, Italy; 5Italian Ministry of Education, 88900 Crotone, Italy; 6Department of Didactics of Musical, Plastic and Body Expression, Faculty of Teaching Training, University of Extremadura, 10071 Cáceres, Spain

**Keywords:** body dissatisfaction, height dissatisfaction, mental health, psychometrics, factor analysis, figural scales, contour line drawings

## Abstract

Body dissatisfaction (BD) is an important public health issue as it negatively influences the physical and psychosocial wellbeing of children/early adolescents. Available measures of BD for this population are scarce, have a significant bias, or only evaluate weight-related dissatisfaction. This study, through the exploratory factor analysis (EFA), aims to develop and validate the Italian (Study 1) and Spanish (Study 2) versions of a new tool, the Body Image Bidimensional Assessment (BIBA), which is not subject to sex–age–race biases and is able to identify BD related to weight and height among children/early adolescents. Study 3 regards the confirmatory factor analysis (CFA), testing the measurement of invariance across sex and country. The BIBA has a two-factor structure (i.e., weight and height dissatisfaction) according to studies 1 and 2. McDonald’s ω ranged from 0.73 (weight) and 0.72 (height) with good reliability. CFA confirmed the two-factor model as a good fit for the Italian and Spanish samples. Finally, partial metric and scalar invariance of the BIBA dimensions across sexes and nations emerged. The BIBA has proven to be an easy-to-use tool that identifies two BD dimensions among children/early adolescents who could benefit from prompt educational interventions.

## 1. Introduction

Body dissatisfaction refers to negative feelings, thoughts, and attitudes towards one’s own body image. Body image is a multifaceted construct related to the feelings and emotions that a person experiences regarding how he/she perceives and feels with and within his/her own body (i.e., physical appearance, shape, and size) and the behaviors associated with such experiences. The development of the body image, which begins at birth, is correlated to the social, cognitive, and personal development of the person [1] and is largely influenced by the environment (e.g., family, peers, etc.) and beauty models proposed by society [2,3], among others.

Recent research has demonstrated that preschool children’s beauty ideals may not match their own body image [4]. Although wanting a different body is not always synonymous with body dissatisfaction, poor body image can appear at a very early age of development [4,5]. Body dissatisfaction may not only prevent children and teenagers from adopting healthy behaviors (e.g., physical activity or sports) but is also a risk factor for unsafe behaviors (e.g., dieting and fasting) [6], poor interpersonal relations, and mental health problems (e.g., eating disorders) [7].

Prevention programs targeted to children (ages 5–18) appear complex and have limited effectiveness on specific psychological symptoms. However, the best prevention interventions focus on risk rather than problematic behaviors [8]. Thus, the early recognition of risk factors like body dissatisfaction can be a successful strategy for preventive public health.

Research has shown that it is necessary to use specific instruments for the child and adolescent population [9]. Until now, existing measures have focused on assessing body dissatisfaction in relation to weight [10,11,12,13] without considering that for many children, especially the youngest, height could be a measure of greater concern in relation to their self-image [14]. Height can be a source of body dissatisfaction for some children, particularly if they feel that they are taller or shorter than their peers. In fact, although limited, empirical evidence has demonstrated that shorter people or those dissatisfied with their body height tend to develop a lower level of self-esteem and have poorer mental health compared to their taller counterparts [15]. Height development in children is determined by a number of factors, including genetics, nutrition, and overall health. However, the exact timeline and rate of growth can vary widely from child to child. Therefore, it would be important to be able to measure body dissatisfaction relative to one’s own height.

In addition, the measures of body dissatisfaction are biased by the sex, age, or race of the stimuli presented as they represent Caucasian boys and girls with clearly defined physiognomic traits [12,13,16,17].

Moreover, available tools do not provide quantitative estimates of the level of body dissatisfaction [12,13].

For these reasons, an easy-to-use, valid, and effective tool for children that does not suffer from sex-age-race biases and is able to identify more body dissatisfaction dimensions in large screening settings is needed.

To fill this gap, we created the Body Image Bidimensional Assessment (BIBA) based on the Body Image Dimensional Assessment (BIDA), which measured body dissatisfaction in adults [18,19]. The BIBA uses the same kind of images, but the rating scale is composed of letters to allow an easier comprehension of the test for children who do not know the decimal number system. The BIBA is a questionnaire that employs neutral figures as items to evaluate the subjective and affective dimensions of body image. This self-administered questionnaire consists of four questions to be answered in relation to six increasingly large images (i.e., in weight or height) using a scale ranging from the letter A to the letter F. The figures are schematic and do not relate to any specific sex, ethnicity, or age.

Based on the above, the first aim of the current study was to develop and validate the Italian and Spanish versions of the BIBA in samples of children/early adolescents between 6–10 and 11–14 years old, respectively. Achieving this goal would ensure that the BIBA could be reliably and validly used in these countries. The second aim of the present study was to investigate whether this measurement displays measurement invariance across sexes and countries. Measurement invariance assesses the psychometric equivalence of a construct across groups or measurement occasions. It demonstrates that a construct has the same meaning to those groups or across repeated measurements [20]. Our hypothesis is that body dissatisfaction between the ages of six and thirteen is related to not only one’s own weight, but also one’s height.

## 2. Study 1: Factor Structure and Internal Consistency of the BIBA in the Italian Sample

### 2.1. Participants

The study was conducted by the researchers of the Outpatient Unit for Clinical Research and Treatment of Eating Disorders and the Department of Medical and Surgical Sciences of the University Magna Graecia of Catanzaro.

According to the indications of the regional school office, the sample was selected to be as representative as possible of the population, including primary public school children and secondary public school adolescents from rural and urban areas. An information letter was sent to each of the pupils’ parents detailing the purpose and procedures of the study and asking for written consent for their child’s participation. Only those children whose parents gave prior consent were enrolled. There were no specific inclusion criteria other than age (6 to 13 years) and desire to participate in the study. The only exclusion criterion was the lack of parental consent or the student’s unwillingness to participate. The overall participation rate was 91%. Thus, from February 2016 to December 2017, we recruited a non-clinical sample of 478 children aged between 6 and 13 years in 7 primary education centers in Calabria.

An informed consent was signed by the parents of the children after a detailed description of the research before any procedure took place.

### 2.2. Measures

Body Image Bidimensional Assessment (BIBA).

This self-administered questionnaire consists of eight questions: the first four refer to a panel illustrating six figures (identified with the letters from A to F) of increasing weight; successively, the same four questions are proposed in relation to the second panel of six figures of increasing height (also identified with the letters from A to F, Figure 1). The figures are schematic so as to not be related to any specific sex, ethnicity, or age to ensure that the observer’s attention is focused only on body shape/height. The four questions to answer about each panel of figures (i.e., weight and height) are:

Q1: What do you currently look like? *(Come sei ora?)*

Q2: What would you like to look like? *(Come ti piacerebbe essere?)*

Q3: Which is the most beautiful figure? *(Quale è la figura più bella?)*

Q4: What do most children of your own sex and age look like? *(Come sono la maggior parte dei bambini/e del tuo stesso sesso ed età?)*

These questions emerge from the clinical experience with patients with EDs or body image dissatisfaction. Three indexes are calculated to measure body dissatisfaction:


In relation to the weight:


Dissatisfaction with ideal weight (DIW): expresses the difference between the actual weight and ideal weight.

Dissatisfaction with weight canon (DWC): expresses the difference between the current weight and the weight of the most beautiful figure.

Comparative weight dissatisfaction (CWD): expresses the difference between the current weight and the weight of peers.


In relation to the height:


Dissatisfaction with ideal height (DIH): expresses the difference between the actual height and ideal height.

Dissatisfaction with height canon (DHC): expresses the difference between the current height and the height of the most beautiful figure.

Comparative height dissatisfaction (CHD): expresses the difference between the current height and the height of peers.

A numerical rank transformation is performed for each response with an increasing value from 1 to 6 (A = 1, B = 2, C = 3, D = 4, E = 5, F = 6). The formulas to calculate each index are:DIW = (Q1 − Q2) × 100/5(1)
DWC = (Q1 − Q3) × 100/5(2)
CWD = (Q1 − Q4) × 100/5(3)
DIH = (Q1 − Q2) × 100/5(4)
DHC = (Q1 − Q3) × 100/5(5)
CHD = (Q1 − Q4) × 100/5(6)

### 2.3. Sociodemographics

After completing the BIBA, researchers measured each participant’s weight and height; the body mass index (BMI) was calculated (BMI= [weight (kg)/height (cm) * height (cm)] × 10,000) and transformed according to the criteria of Cole (>95th percentile = overweight; 85th> <95th percentile = risk of overweight; 5th> <85th percentile = normal weight; <5th percentile = underweight) [21], taking into consideration each participant’s sex and age.

### 2.4. Statistical Analysis

Statistical analyses were performed with R [22]. Socio-demographics, presented as means, standard deviations (SD), frequencies, percentages, and differences according to sex and BMI categories were tested through a chi-square test. Analyses were run among the raw calculated indexes of the BIBA (i.e., DIW, DWC, CWD, DIH, DHC, and CHD).

Exploratory factor analysis (EFA) was performed using the R *psych* package [23]. EFA with a principal axis factoring method was chosen to investigate the factor structure of the BIBA. We used the Kaiser–Meyer–Olkin (KMO) test and Bartlett’s test to determine whether the data were suited for EFA. A parallel analysis method was used to determine the number of factors to extract. After the extraction, we applied an oblimin rotation because it was not tenable to assume that the extracted factors were orthogonal. Factor loadings > |0.40| were considered meaningful.

### 2.5. Results

After eliminating missing data, the final sample comprised 415 children (226 females) aged between 6 and 13 years (mean age = 9.82, SD = 2.01). The frequency of boys and girls was homogeneous through age (*χ*^2^ = 11.513, df = 7, *p* = 0.118) and BMI categories (*χ*^2^ = 7.453, df = 3, *p* = 0.144).

#### Factor Structure of the BIBA

Both the KMO test (0.67) and Bartlett’s test (*χ*^2^ (15) = 744.57, *p* < 0.001) indicated that the data were suitable for EFA. The parallel analysis suggested a two-factor solution (Figure 2) that explained 55.4% of the variance (factor 1 = 28.9%; factor 2: 26.5%).

Factor 1 consisted entirely of weight-related variables and factor 2 consisted of height-related variables. (Table 1).

## 3. Study 2: Factor Structure and Internal Consistency of the BIBA in the Spanish Sample

This study followed the same methods and procedures as in Study 1.

### 3.1. Participants

Researchers from the Department of Didactics of Musical, Plastic, and Body Expression of the University of Extremadura recruited a sample of 417 children (221 females) aged between 6 and 13 years (mean age = 9.50, SD = 1.57) in five public education centers in Extremadura. The overall participation rate was 93%.

The frequency of boys and girls was homogeneous through age (*χ*^2^ = 3.670, df = 6, *p* = 0.721) and BMI categories (*χ*^2^ = 0.643, df = 3, *p* = 0.887).

### 3.2. Measures

The Spanish version of the BIBA (Figure 3) was simultaneously translated by the main author of the Italian version. The Spanish questions are:

Q1: What do you currently look like? *(¿Cómo eres ahora?)*

Q2: What would you like to look like? *(¿Cómo te gustaría ser?)*

Q3: Which is the most beautiful figure? *(¿Cuál es la figura más bonita?)*

Q4: What do most children of your own sex and age look like? *(¿Cómo son las mayor parte de los niños/as de tu mismo sexo y edad?*)

### 3.3. Results

#### Factor Structure of the BIBA

The Kaiser–Meyer–Olkin coefficient (KMO = 0.67) and the Bartlett test (*χ*^2^ (15) = 679.01, *p* < 0.001) showed that the data were appropriate for EFA. The parallel analysis suggested a two-factor solution (Figure 4) that explained 53.1% of the variance (factor 1 = 28.3%; factor 2: 24.8%).

Factor 1 was made up entirely of weight-related variables and factor 2 of those related to height, as in Study 1 (Table 2).

## 4. Study 3: Confirmatory Factor Analysis and Measurement of Invariance of the BIBA across Countries and Sexes

### 4.1. Participants and Methods

The last sample comprised 869 children: 452 Italian (232 female and 219 male) and 415 Spanish (193 female and 222 male) participants aged between 6 and 13 years (Italian sample: mean age = 9.85, SD =1.94; Spanish sample: mean age = 9.50, SD = 1.52) recruited by the same researchers in the same centers of studies 1 and 2.

Initially, we performed a confirmatory factor analysis (CFA) of each sample to test the unidimensional and bidimensional structure of the BIBA using the R packages “*semTools*” [24] and “*lavaan*” [25]. To assess the goodness of the model fit, we used the following indices: χ^2^, Root Mean Square Error of Approximation (RMSEA), Comparative Fit Index (CFI), Tucker–Lewis Index (TLI), and Standardized Root Mean Square Residual (SRMR). For RMSEA, a value below 0.05 indicated a good fit, a value between 0.05 and 0.08 represented a fair fit, and a value between 0.08 and 0.10 signified a mediocre fit [26,27]. For CFI and TLI, a value above 0.90 was considered acceptable [28]. For SRMR, a value below 0.05 was considered an acceptable fit [28,29].

Afterward, we determined the internal consistency through the McDonald’s ω total, as Cronbach’s α has strict limitations [30,31,32]. We also conducted a multiple-group CFA to determine the measurement invariance of the BIBA across sexes (men and women) and countries (Italy and Spain). The process was carried out in three stages. First, the configural invariance model was established as a baseline to check if the factor structure was consistent across the groups. Second, the metric invariance model was established to determine if the factor loadings for the items were the same for both sex and country. This is crucial for meaningful comparisons between groups [33]. Finally, the scalar invariance model was established to check if the factor loadings and intercepts were the same across the groups. Since the three models were nested within each other, measurement invariance was determined based on the overall model fit and changes in fit indices between the models. Measurement invariance is considered to be supported if the comparison between the two models meets the following criteria: a non-significant Δχ2, ΔRMSEA < 0.050, ΔCFI < 0.004, and ΔSRMR ≤ 0.01 [34].

To check the source of the lack of equivalence, R modification indices were also investigated. In this regard, when a constraint is untenable, it can be relaxed to obtain partial invariance [35].

### 4.2. Results

#### 4.2.1. CFA in the Italian Sample

According to the fit indices, the two-factor model with correlated latent variables (Figure 5a) showed the best fit, demonstrating a very good model fit (CFI = 0.984, TLI = 0.966, RMSEA = 0.061, relative chi-square (χ^2^/df) = 2.30, and SRMR = 0.041) compared to the unidimensional model (CFI = 0.57, TLI = 0.29, RMSEA = 0.28, relative chi-square (χ^2^/df) = 36.67, and SRMR = 0.17).

The internal consistency of the BIBA was acceptable with a McDonald’s ω_t_ of 0.77 and 0.76 for weight and height factors, respectively.

#### 4.2.2. CFA in the Spanish Sample

Following the previous results of the Italian sample, we tested a two-factor model. We found an excellent fit: (CFI = 0.98, TLI = 0.97, RMSEA = 0.05, relative chi-square (χ^2^/df) = 2.69, and SRMR = 0.045, Figure 5b).

The internal consistency of the BIBA was acceptable with a McDonald’s ω_t_ of 0.73 and 0.72 for weight and height factors, respectively.

#### 4.2.3. Measurement of Invariance across Countries

Multiple-group CFA was performed to examine the measurement invariance across countries. Fit indices for the three models and the differences between the pairs of nested models are displayed in Table 3.

First, to confirm the configural invariance of the BIBA, an unconstrained analysis was conducted on two country groups, which showed a good fit for the configural invariance model (M1) (χ^2^ = 34.96, RMSEA = 0.06, CFI = 0.98, TLI = 0.97, SRMR = 0.04).

Second, for the metric invariance analysis, the factor loadings were constrained to be equal between the Italian and Spanish groups, but this resulted in a worse fit. However, after relaxing the constraint on the “Dissatisfaction with ideal weight” item, partial metric invariance (M2) was achieved. The results showed no significant difference in Δχ2 (Δχ^2^  =  9.963, *p*  =  0.08) and minimal changes in CFI, TLI, RMSEA, and SRMR (ΔCFI = 0.004; ΔTLI = −0.003; ΔRMSEA = 0.003, and ΔSRMR = −0.011), indicating that the partial metric invariance of the BIBA was consistent across countries.

Third, the scalar invariance analysis was then conducted by constraining the factor loadings and intercepts of the items to be equal across the two countries. The results showed that the model fit worsened, but after relaxing the constraint on the threshold of the “Dissatisfaction with ideal height” item, a partial scalar invariance model (M3) was achieved. The changes in CFI, TLI, RMSEA, and SRMR (0.003, −0.001, 0.000, −0.002) were smaller than the recommended cutoff values for rejecting measurement invariance, and Δχ^2^ was not significant (Δχ^2^ = 6.62, *p*  =  0.09), indicating that the factor loadings and item intercepts were invariant for both Italian and Spanish groups. This demonstrated the satisfaction of scalar invariance.

#### 4.2.4. Measurement of Invariance across Sex

Multiple-group CFA was run to examine the measurement invariance across sex. Fit indices for the three models and the differences between the pairs of nested models are displayed in Table 4.

First, the configural invariance was assessed by estimating both sex groups without equality constraints. The results confirmed the configural invariance of the BIBA (M1) as indicated by the fit indices (χ^2^ = 36.666, RMSEA = 0.061, CFI = 0.983, TLI = 0.963, SRMR = 0.034).

Then, metric invariance was achieved by constraining the factor loadings to be the same between male and female groups, but this solution worsened the model fit.

An inspection of the modification indices revealed a constraint that was not tenable (factor loadings on the item “Dissatisfaction with weight canon”). After it was relaxed, a partial metric invariance model was achieved. Compared with M1, the partial metric invariance model (M2) reported that Δχ^2^ was not significant (Δχ^2^  =  1.193, *p*  =  0.95) and value changes of CFI (ΔCFI), TLI (ΔTLI), RMSEA (ΔRMSEA), and SRMR (ΔSRMR) were −0.003, −0.014, 0.013, and −0.003, respectively. These results showed that the partial metric invariance of the BIBA held across sex.

Finally, the scalar invariance was assessed by restricting factor loadings and interceptions of items to make them equal between the two sexes. Results from the scalar invariance model (M3) showed that the model worsened the fit. An inspection of the modification indices revealed a constraint that was not tenable (threshold of item “Comparative height dissatisfaction”), but a partial scalar invariance model was achieved after it was relaxed. In fact, compared with M2, the values of ΔCFI, ΔTLI, ΔRMSEA, and ΔSRMR were all smaller than the recommended cutoff values for rejecting measurement invariance (0.000, −0.003, 0.003, −0.001), and Δχ^2^ was not significant (Δχ^2^  =   3.395, *p*  =  0.33).

## 5. General Discussion

The objective of the present research was to create and validate the Italian (Study 1) and the Spanish (Study 2) versions of the Body Image Bidimensional Assessment (BIBA) and to evaluate whether the BIBA exhibited measurement invariance regarding sex and country (Study 3). The aim was to create a tool that is simple for children and adolescents to use and is both reliable and effective in measuring body dissatisfaction, covering not only dissatisfaction with weight, but also with height.

The results of the exploratory (studies 1 and 2) and the confirmatory factor analyses (Study 3) supported a final six-indexes and two-factor structure of the BIBA. The model accounted for more than half of the variance and demonstrated good model fit in both the Italian and Spanish samples.

Our hypothesis was strongly supported by the striking finding that the BIBA factor structure validated body height dissatisfaction as a unique dimension of body dissatisfaction, similar to weight dissatisfaction. Prior research on body dissatisfaction had primarily emphasized weight and appearance dissatisfaction, rather than height dissatisfaction. However, empirical data have revealed that individuals who are shorter or unhappy with their height tend to experience reduced levels of self-esteem and confidence, along with poorer mental health outcomes, compared to taller peers [15]. Moreover, a recent study has indicated that while height dissatisfaction may not predict social anxiety or support, it can anticipate feelings of loneliness among adolescents [33].

The BIBA introduces a novel feature that also serves as a key advantage: the utilization of poorly structured stimuli, as in the case of projective assessment techniques, not connected to age, sex, or race, that may facilitate freer expression of children.

The measurement of invariance across countries, the results of the single group CFA, and the configural invariance analysis revealed that the BIBA assessed the same construct across Italy and Spain. However, full metric and scalar invariance were not supported, and only partial invariance was achieved across nations. Specifically, the item “DIW,” which measures the difference between actual and ideal weight, was non-invariant, indicating that Italian and Spanish children perceived this item differently. The Italian sample showed a higher DIW loading than the Spanish sample, which could be due to the higher prevalence of obesity and severe obesity among Spanish children than Italian children [36], as well as the greater reduction in obesity rates observed in Italy compared to Spain [37]. These findings suggest that the Spanish may have a stronger internalization of the body dissatisfaction construct compared to the Italian sample.

Our findings indicate that the item “DIH” was non-scalar invariant, as the Italian sample had a higher intercept value for this item compared to the Spanish sample. This outcome could be explained by recent research that suggests that the average height of Spanish children and adolescents, particularly males, is greater than that of Italians [38].

The present study has found partial evidence to support the metric and scalar invariance between males and females. In terms of metric invariance, one particular item, “DWC,” which refers to the discrepancy between one’s current weight and the weight of the ideal figure, showed non-invariance. Specifically, our findings indicated that the female participants reported a higher loading on the “DWC” item compared to the male participants. This suggests that girls may experience more distress in relation to the societal beauty standards promoted by mass culture, resulting in higher scores on the “DWC” item.

The results of our study provide support for the hypothesis that girls are more susceptible to the pressures of conforming to societal beauty standards. In fact, prior research has shown that girls who experience greater body dissatisfaction and engage in dietary restraint are more likely to live within a subculture that supports the thin ideal and encourages dieting [38].

Overall, our findings contribute to the growing body of literature that highlights the impact of societal beauty standards on individuals’ body image and mental health. Our study’s partial support for metric and scalar invariance between sexes underscores the need for further research to better understand the unique experiences and challenges faced by different sex groups in relation to body image and self-esteem.

An earlier study revealed that a higher weight status in children can potentially lead to lower peer competence over time. Interestingly, the impact of weight on peer competence seems to affect girls at a much earlier age than boys, as reported in the study [39]. These findings may suggest that the effects of weight status on peer relationships can be more severe for girls than boys, leading to social exclusion and a lower sense of belonging.

In our study, we found that boys exhibited a higher “CHD” intercept value than girls. This finding is consistent with the well-established association between height and male gender roles. Taller men tend to be perceived as more dominant and assertive, traits that are highly valued in traditional male gender roles [40]. Height is therefore a central component of male body image and plays a significant role in the development of masculine identity [41].

It is worth noting that the relationship between height and gender roles is complex and multifaceted. While height may be associated with traditionally masculine traits, it is important to recognize that masculinity is a socially constructed concept that varies across cultures and over time. Therefore, this result highlights the importance of considering sex and gender-specific factors in the development of body image. The result also suggests the need for further research to better understand the complex relationship between height, sex and gender roles, and masculine/feminine identity, and how these factors may affect body image.

It is important to note that this study has a limitation: it does not provide normative data about the Body Image Bidimensional Assessment (BIBA). On one hand, the authors did not identify any established measure that could be used as a “golden standard” for the estimation of the cut-off; on the other, the sample sizes in the study were not large enough to permit stratification for various demographic variables such as country, sex, age, and BMI, which would have allowed for the calculation of z scores. Therefore, further research is necessary to estimate normative data for the BIBA, which would enable the determination of cut-off points.

## 6. Conclusions

The BIBA has been shown to be a reliable and user-friendly tool in measuring multiple dimensions of body dissatisfaction in children and young adolescents. It has a two-factor structure, specifically relating to weight and height dissatisfaction, and its measurement invariance has been confirmed.

This research has important implications for both researchers and healthcare professionals, as it enables direct comparisons of more dimensions of body dissatisfaction across sex and two countries.

The results highlight the potential benefits of using the BIBA to detect body dissatisfaction in children and adolescents, as early intervention can help prevent negative health outcomes and mental health problems associated with body dissatisfaction.

## Figures and Tables

**Figure 1 ijerph-20-03626-f001:**
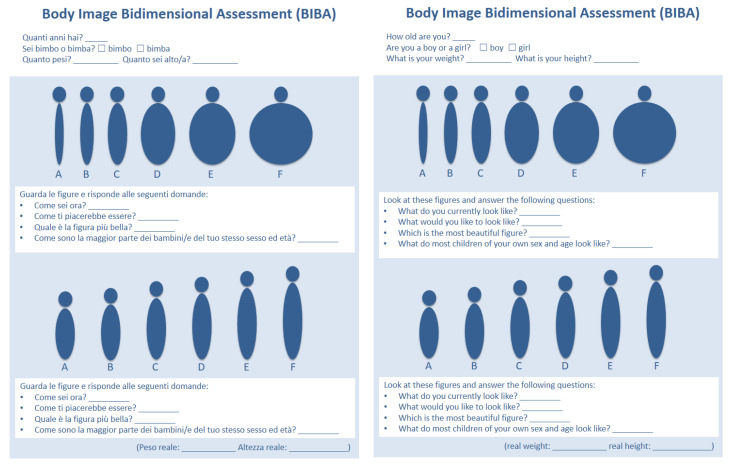
The BIBA Italian version.

**Figure 2 ijerph-20-03626-f002:**
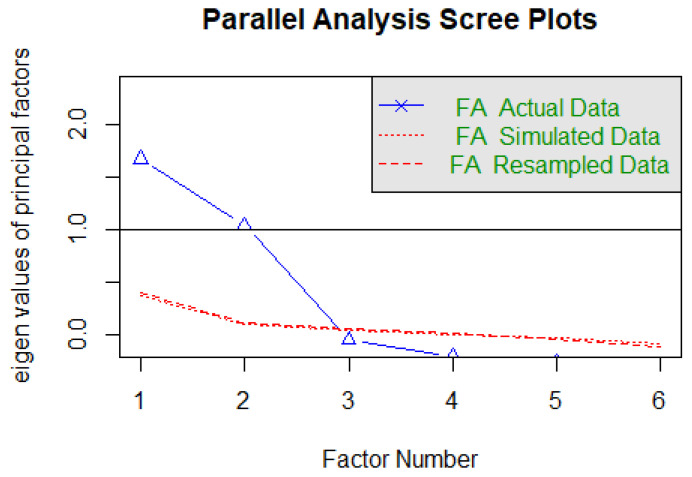
The parallel analysis of the BIBA Italian version.

**Figure 3 ijerph-20-03626-f003:**
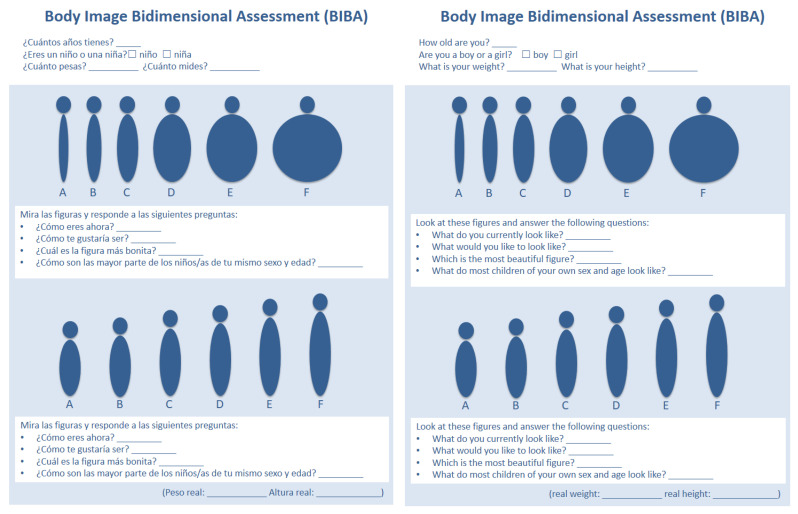
The BIBA Spanish version.

**Figure 4 ijerph-20-03626-f004:**
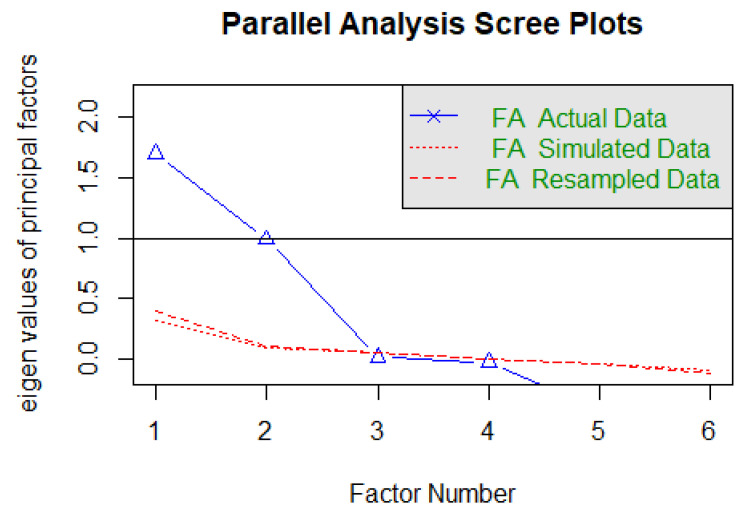
The parallel analysis of the BIBA Spanish version.

**Figure 5 ijerph-20-03626-f005:**
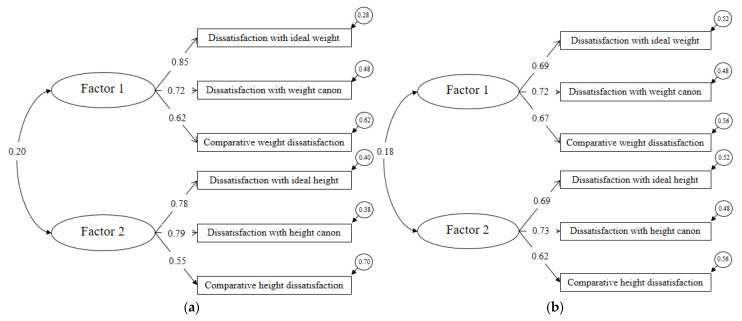
(**a**) Confirmatory factor analysis of the Italian BIBA; (**b**) confirmatory factor analysis of the Spanish BIBA.

**Table 1 ijerph-20-03626-t001:** Italian EFA: matrix of rotated components.

	Components	
	1	2
Dissatisfaction with ideal weight	0.895	
Dissatisfaction with weight canon	0.732	
Comparative weight dissatisfaction	0.597	
Dissatisfaction with ideal height		0.814
Dissatisfaction with height canon		0.776
Comparative height dissatisfaction		0.556

**Table 2 ijerph-20-03626-t002:** Spanish EFA: matrix of rotated components.

	Components	
	1	2
Dissatisfaction with ideal weight	0.814	
Dissatisfaction with weight canon	0.683	
Comparative weight dissatisfaction	0.748	
Dissatisfaction with ideal height		0.698
Dissatisfaction with height canon		0.778
Comparative height dissatisfaction		0.624

**Table 3 ijerph-20-03626-t003:** Fit indices for measurement invariance tests for countries.

	Robust Model Fit Indices							Model Difference					
Model	χ^2^	df	CFI	TLI	RMSEA	SRMR	ΔM	Δ χ^2^	Δdf	ΔCFI	ΔTLI	ΔRMSEA	ΔSRMR
M1	34.960	14	0.98	0.97	0.06	0.04							
M2 *	44.923	19	0.98	0.97	0.06	0.05	M2 VS. M1	9.963	5	0.004	−0.003	0.003	−0.011
M3 *	51.538	22	0.98	0.97	0.06	0.05	M3 VS. M2	6.615	3	0.003	−0.001	0.000	−0.002

M1: configural invariance; M2 *: partial metric invariance; M3 *: partial scalar invariance. All the Δ χ^2^ were not significant.

**Table 4 ijerph-20-03626-t004:** Fit indices for measurement invariance tests for sex.

	Robust Model Fit Indices							Model Difference					
Model	χ^2^	df	CFI	TLI	RMSEA	SRMR	ΔM	Δ χ^2^	Δdf	ΔCFI	ΔTLI	ΔRMSEA	ΔSRMR
M1	36.666	14	0.98	0.96	0.06	0.03							
M2 *	37.859	19	0.96	0.98	0.05	0.04	M2 VS. M1	1.193	5	−0.003	−0.014	0.013	−0.003
M3 *	41.255	22	0.99	0.980	0.05	0.04	M3 VS. M2	3.395	3	0.000	−0.003	0.003	−0.001

M1: configural invariance; M2 *: partial metric invariance; M3 *: partial scalar invariance. All the Δ χ^2^ were not significant.

## Data Availability

The data presented in this study are available on request from the corresponding author.

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
