# Peer review of "Development, Validation, and Measurement Invariance of the Body Image Bidimensional Assessment (BIBA) in Italian and Spanish Children and Early Adolescent Samples"

_ijerph, 2023, doi:10.3390/ijerph20043626_

Round 1

Reviewer 1 Report

This paper proposes a new tool called BIBA and validates in children and early adolescents. There are some major and minor concerns which must be addressed. 

Ethical approval is necessary. Although informed consent was acquired, it was not stated whether the study was approved by the institutional review board or not. Was consent obtained directly from the subjects or their guardians? This study dealt with minors. Please clarify these issues.

The terms “sex” and “gender” were used interchangeably. However, these two are two distinct entities. The term sex reflects the biological entity, while gender refers to the socially constructed entity. Which definition was used in this study? Please stick to one definition only throughout the manuscript. 

Specific comments

  • Line 49: What recent research? Please add a reference.

  • Line 61: Which existing measurements?

  • Heading of section 2 (materials and methods) is missing

  • Line 226-227: Which previous studies?

  • Line 279: Rephrase to “measurement of invariance”

  • Line 325: Delete redundant spaces after the comma

  • Line 376: I am afraid the Institutional Review Board statement is inevitable here. 

Author Response

This paper proposes a new tool called BIBA and validates in children and early adolescents. There are some major and minor concerns which must be addressed. 

Answer: We are grateful for the reviewer’s issues. We will try to address and answer to all reviewer’s comments. 

Ethical approval is necessary. Although informed consent was acquired, it was not stated whether the study was approved by the institutional review board or not. Was consent obtained directly from the subjects or their guardians? This study dealt with minors. Please clarify these issues.

Answer: Thank you for this comment. We added this information at the end of the manuscript. 

The terms “sex” and “gender” were used interchangeably. However, these two are two distinct entities. The term sex reflects the biological entity, while gender refers to the socially constructed entity. Which definition was used in this study? Please stick to one definition only throughout the manuscript. 

Answer: Thank you very much for this comment, you are right. We have modified through the text. In fact, we have considered the sex of participants.

Specific comments

  • Line 49: What recent research? Please add a reference.

Answer: We added a reference. Thank you!

  • Line 61: Which existing measurements?

Answer: We added a reference. Thank you!

  • Heading of section 2 (materials and methods) is missing

Answer: The manuscript was divided into three studies. Study 1 (Italian validation), Study 2 (Spanish validation) and Study 3 (Measurement invariance). Each study was divided into several sections: participants, methods, statistical analysis and results for a better comprehension of the potential reader.  

  • Line 226-227: Which previous studies?

Answer: Thank you, we added a reference. 

  • Line 279: Rephrase to “measurement of invariance”

Answer: Thank you for this comment, we rephrased the sentence.  

  • Line 325: Delete redundant spaces after the comma

Answer: Thank you. It was a typo error. 

  • Line 376: I am afraid the Institutional Review Board statement is inevitable here. 

Answer: Thank you for this suggestion. We have added it.  

Reviewer 2 Report

Overview

The authors aimed to develop and validate the Italian and Spanish versions of the Body Image Bidimensional Assessment in samples of children/early adolescents and investigate whether this measurement displays measurement invariance across genders and countries.

The topic is very interesting.

Below are my comments.

Specific comments

Keywords

To optimize the search for your manuscript through search engines, I recommend using keywords other than the title. The words in the title are already keywords. Replace the words "body image; measurement invariance; children; adolescents" with other words than the title.

Introduction

In the introduction the authors did a good job of synthesizing the literature; the aims and hypothesis are clear.

Methods

The methodology is clearly explained.

The statistical techniques used are appropriate.

The exploratory factor analysis and the confirmatory factor analysis were done correctly.

Results

The results are written correctly.

The figures and tables are explanatory.

Discussion

The discussions are clear and to the point.

The limitations are described.

Conclusions

The authors' conclusions are justified.

The take-home message is clear.

Author Response

We want to thank the reviewer for his/her positive comments.

  • "To optimize the search for your manuscript through search engines, I recommend using keywords other than the title. The words in the title are already keywords. Replace the words "body image; measurement invariance; children; adolescents" with other words than the title."

Answer: Thank you for this comment. We have followed your suggestion. 

Reviewer 3 Report

This paper developed and validated a effective tool, which does not suffer from sex-age-race biases, and able to identify body dissatisfaction related to weight and height among children/early adolescents. It is conducive to early identification of children/early adolescents' body dissatisfaction and early prevention of unsafe behaviors and mental health problems. Some suggestions are as follows:

1. Abstract part: Can the contents of lines 28-30 and 31-33 be merged

2. Introduction part: It is suggested to add some literature or evidence about the impact of height dissatisfaction on mental health in lines 61-65.

3. Line 96 Are there criteria for the inclusion of participants, or are children who have the will to participate included?

4. At the end of the discussion, it is suggested to add the idea of education and intervention measures for children with physical dissatisfaction screened out (optional).

Author Response

This paper developed and validated an effective tool, which does not suffer from sex-age-race biases, and able to identify body dissatisfaction related to weight and height among children/early adolescents. It is conducive to early identification of children/early adolescents' body dissatisfaction and early prevention of unsafe behaviors and mental health problems. Some suggestions are as follows.

Answer: Thank you for your positive evaluation of our manuscript. We have followed your suggestion. We hope that our answers will adequately satisfy your concerns and that you can find it improved enough to be accepted by IJERPH for publication.

  • Abstract part: Can the contents of lines 28-30 and 31-33 be merged?

Answer: Thank you for this comment. We have tried to rephrase the contents of those sentences.

  • Introduction part: It is suggested to add some literature or evidence about the impact of height dissatisfaction on mental health in lines 61-65.

Answer: Thank you for this suggestion, we have followed it. Unfortunately there is not so much.

  • Line 96: Are there criteria for the inclusion of participants, or are children who have the will to participate included?

 Answer: As participants were all minor, we sent a letter to their parents to explain the aims and procedures of the study. We did not set any particular criteria for the inclusion in the study but just the willing to participate and the authorization of parents. Only those children whose parents signed the consent were included.

  • At the end of the discussion, it is suggested to add the idea of education and intervention measures for children with physical dissatisfaction screened out (optional).

Answer: This is a good idea! We have written as follows:” Given the risk of unhealthy behaviors or mental health problems secondary to body dissatisfaction, these findings inform about the potential use of the BIBA to capture body dissatisfaction among children/adolescents who could benefit from early educational interventions.

Reviewer 4 Report

The detailed manuscript review report is attached below.

Author Response

The study aims to standardize a new tool for measuring body dissatisfaction on Italian and Spanish samples. The study denotes the positive efforts of the authors to develop a measure of body dissatisfaction, hitherto little researched construct. My observations are presented below:

(1) The study title is good but I suggest following the same order as written in its abbreviation, BIBA, of the scale terms i. e., Body Image Assessment Bidimensional.

Answer: It was a typo error. We have corrected it.

(2) Similarly, the abstract needs minor amendments. Some implications of the study findings will make readers understand its utility.

Answer: Thank you very much for this suggestion. We have modified the abstract accordingly.

(3) a)The introduction also needs modifications. The very first sentence needs corrections. I suggest defining the key term of body dissatisfaction in the first paragraph.b) Some more relevant measures of body dissatisfaction should ne surveyed and their conclusions should be added. Based on their weaknesses, the authors should put their research question to develop a new scale on body dissatisfaction. The authors should add reviews on adolescent age group. The authors are advised to improve the study arguments. There is no mention study. C)The source of the hypothesis, body dissatisfaction between the ages of 6 and 13 is related to not only one's own weight but also one's height, should be mentioned clearly. D)There is sufficient age range of the participants and there may be significant gap in the 2 understanding of body image related issues. How authors have addressed this difference? The authors should clearly mention the age ranges of children and early adolescents.

Answer: We thank the reviewer for sharing his/her point of view.

  1. We agree with defining body dissatisfaction in the first paragraph as follows: “Body dissatisfaction refers to negative feelings, thoughts and attitudes towards own body image. Body image is a multifaceted construct related to the feelings and emotions that the person experiences regarding how he/she perceives feels with and within his/her own body (i.e., physical appearance, shape and size) and the behaviors associated with such experiences.
  2. We made a deep review of existing measures of BD for children but we decided not to make a list with the biases of each one, but rather a summary of the problems that the existing measures present in general. The reviewer can find these measures as references.
  3. You are right. we have rephrased as follows in the introduction: “In fact, although limited, empirical evidence has demonstrated that shorter people or those dissatisfied with their body height tend to develop a lower level of self-esteem and have poorer mental health compared to their taller counterparts [15]. Height development in children is determined by a number of factors, including genetics, nutrition, and overall health. However, the exact timeline and rate of growth can vary widely from child to child. Height can be a source of body dissatisfaction for some children, particularly if they feel that they are taller or shorter than their peers. Therefore, it would be important to be able to measure body dissatisfaction relative to one's own height.”
  4. Again, the reviewer is right. Considering the gap in the literature and the pre-existing tool to measure body dissatisfaction in adults (BIDA), we thought that it would be useful to create a test for the infant-juvenile age. We think that the childhood age ranges from 6 to 10 years old, and the first youth age from 11 to 14. Adolescence typically begins around the age of 11 and lasts until the late teens or early twenties so we decided to explore the early adolescence. Of course, the exact timing can vary depending on the individual, but these the frames generally accepted. We have added the age in the text as suggested.

(4) The methods section also needs improvements. It should start with the research design used and the sampling method employed. The authors should give the reasons as to why participants with a long age range (7 to 13 years) were chosen. They should also mention the participants’ recruitment method. Moreover, the inclusion and exclusion criteria are absent.

Answer: Thank you for this suggestion. We have included this information: “According to the indications of the regional school office, the sample was selected to be as representative as possible of the population including primary public school children and secondary public school adolescents from rural and urban areas. An information letter was sent to each of the pupils' parents detailing the purpose and procedures of the study and asking for written consent for their child's participation. Only those children whose parents gave prior consent were enrolled. There were no specific inclusion criteria other than age (6 to 13 years) and desire to participate in the study. The only exclusion criterion was the lack of parental consent or the student's unwillingness to participate. The overall participation rate was 91%.”

(5) The results section also needs improvements. The authors are suggested to add brief discussion of each study separately. They should also present arguments why the next study was needed. How validity (content, criterion and construct) was estimated? The authors should correct the reporting of the statistical values following standard reporting guidelines such as APA. For example, a p-value above .001 is reported as p = .05. Please correct this error throughout the manuscript.

Answer: Thank you for these suggestions. Study 2 was necessary to be able to test the validity of the scale in the Spanish population after having tested it in the Italian sample (Study 1). Study 3 was therefore explained for two reasons: to confirm the results of the first two studies and to verify if the data from each country, taking into account sex and age, were comparable. We have added it in the first paragraph of the discussion.

Regarding the reporting of the statistical values, even if we prefer the APA approach to present the results, we followed the authors’ guidelines. We would not have any problem to follow your suggestion but in order to make clear the significance of the results of EFA and CFA according to methods we decided to present the results as they appear in the analyses. For example, if for “RMSEA, a value smaller than .05 represented a close fit; a value ranging between.05 to .08 indicated a fair fit; a value in the range of .08 to .10 denoted a mediocre fit” it was necessary to specify the p obtained. Thus, we decided to follow the same criteria through the text.

(6) The discussion section needs significant improvements. The discussion should be labelled as general discussion and it should compile findings of all the three studies. The interpretations of the study findings should be done in continuation with the research questions. The author should attempt to interpret the invariance in terms of the impacts of social media and some similar factors. Some theory, research and practice implications of the study findings need addition. Novel findings should be highlighted. Some suggestions for future directions may be added at the end of the discussion.

Answer: Following your suggestions, we have labelled it as “General discussion” and had extended it. We hope you will find this new section improved.

(7) The references are ok.

Answer: Thank you

In essence, the study is a good piece of work and its findings may contribute to the field of body image. Still, small modifications are suggested. I strongly suggest language editing throughout the manuscript. As it stands, the manuscript may be accepted after minor modifications as suggested above. I thank the editor again for the opportunity to review the manuscript.

Answer: Thanks for the time you have dedicated to review the manuscript